# Perioperative Predictive Factors for Tumor Regression and Survival in Non-Small Cell Lung Cancer Patients Undergoing Neoadjuvant Treatment and Lung Resection

**DOI:** 10.3390/cancers16162885

**Published:** 2024-08-20

**Authors:** Fuad Damirov, Mircea Gabriel Stoleriu, Farkhad Manapov, Enole Boedeker, Sascha Dreher, Sibylle Gerz, Thomas Hehr, Evelin Sandner, German Ott, Rudolf Alexander Hatz, Gerhard Preissler

**Affiliations:** 1Department of Thoracic Surgery, Ludwig Maximilian University of Munich (LMU) and Asklepios Lung Clinic, Munich-Gauting, 82131 Gauting, Germany; gabriel.stoleriu@helmholtz-munich.de (M.G.S.);; 2Department of Thoracic Surgery, RBK-Lungenzentrum Stuttgart, Schillerhöhe Lung Clinic, Robert Bosch Hospital, 70376 Stuttgart, Germany; 3Institute for Lung Health and Immunity, Comprehensive Pneumology Center with the CPC-M BioArchive, Helmholtz Center Munich, Member of the German Lung Research Center (DZL), 81377 Munich, Germany; 4Die Radiologie, 80331 Munich, Germany; 5Department of Radiation Oncology, Marienhospital Stuttgart, 70199 Stuttgart, Germany; 6Department of Oncology, Robert Bosch Hospital, 70376 Stuttgart, Germany; 7Department of Clinical Pathology, Robert Bosch Hospital, 70376 Stuttgart, Germany

**Keywords:** lung cancer, PET/CT-scan, lymph node staging, neoadjuvant therapy, pathological response, tumor regression, overall survival

## Abstract

**Simple Summary:**

Lung cancer continues to be the leading cause of cancer-related deaths worldwide. Many patients present advanced disease at the time of the diagnosis. Neoadjuvant therapy aims to reduce the tumor stage and improve the operability of patients, and simultaneously leads to tumor regression. The tumor regression grade reflects the degree of pathological response to therapy. Thus, we conducted this study to identify the predictors for pathologic response after neoadjuvant treatment followed by surgery. The second goal of our research was to find the relationship between survival and tumor regression. Our study revealed that the histology of the primary tumor, lymph node size in the preoperative CT scan (>1.7 cm), and absolute tumor size reduction after neoadjuvant treatment (>2.6 cm) independently predict the effectiveness of tumor regression. Age > 70 years, extended resection > one lobe, and tumor recurrence or metastasis were identified as significant independent predictors of reduced overall survival.

**Abstract:**

Our study aimed to identify predictors for the effectiveness of tumor regression in lung cancer patients undergoing neoadjuvant treatment and cancer resections. Patients admitted between 2016 and 2022 were included in the study. Based on the histology of the tumor, patients were categorized into a lung adenocarcinoma group (LUAD) and squamous cell carcinoma group (SQCA). Ninety-five patients with non-small-cell lung cancer were included in the study. A total of 58 (61.1%) and 37 (38.9%) patients were included in the LUAD and SQCA groups, respectively. Additionally, 9 (9.5%), 56 (58.9%), and 30 (31.6%) patients were categorized with a tumor regression score of I, II, and III, respectively. In multivariable analyses, histology of the primary tumor (SQCA), lymph node size in the preoperative CT scan (>1.7 cm), and absolute tumor size reduction after neoadjuvant treatment (>2.6 cm) independently predict effectiveness of tumor regression (OR [95% confidence interval, p-value] of 6.88 [2.40–19.77, *p <* 0.0001], 3.13 [1.11–8.83, *p =* 0.0310], and 3.76 [1.20–11.81, *p =* 0.0233], respectively). Age > 70 years, extended resection > one lobe, and tumor recurrence or metastasis were identified as significant independent predictors of reduced overall survival. Assessment of tumor size before and after neoadjuvant treatment might help to identify high-risk patients with decreased survival and to improve patient management and care.

## 1. Introduction

Non-small-cell lung cancer (NSCLC) continues to be the leading cause of cancer-related deaths worldwide [1]. Patients with early stages of NSCLC can usually be cured with surgery alone with a 5-year overall survival (OS) rate of 92% in stage IA [2]. Unfortunately, many patients with NSCLC present advanced disease at the time of diagnosis [1]. Patients with locally advanced NSCLC who receive surgery alone show poor OS [2]. Neoadjuvant therapy to reduce the tumor stage, followed by surgery, is the routine treatment of choice in clinical practice for advanced NSCLC [3]. This therapeutic approach aims to reduce the tumor stage, improve operability, and possibly eradicate microscopic metastases, offering a complete curative approach [4]. However, there are no established common standards for neoadjuvant therapy yet [5,6,7]. It is suggested that different neoadjuvant therapy modalities can lead to different tumor regressions. Even though there is no consensus accepted by the WHO for regression grading in NSCLC [8], the degree of remaining viable tumor cells and the extent of fibrosis are crucial prognostic factors [9]. Various regression scoring systems for lung cancer have been suggested, focusing on the proportion of residual tumor cells [10,11,12]. Junker et al. introduced a three-grade regression scheme, where grade I is defined by no or only spontaneous regression, grade IIa represents incomplete regression with more than 10% viable tumor cells, grade IIb denotes incomplete regression with less than 10% viable tumor cells, and grade III signifies complete regression with no viable tumor cells remaining [10,11]. Following these results, the assessment of tumor regression grade (TRG) has emerged as a valuable prognostic indicator in evaluating the efficacy of neoadjuvant therapies [11,12,13]. TRG reflects the degree of pathological response to therapy and holds the potential to guide clinical decisions, improve patient outcomes, and enhance treatment methods [14]. Moreover, the pathologic response without vital tumor cells has been used as a surrogate endpoint in neoadjuvant therapy studies [8,15]. Thus, understanding the complex relationship between neoadjuvant therapy and TRG is critical for optimizing therapeutic approaches and establishing suitable treatment strategies for individual patients.

The cohort of patients receiving neoadjuvant therapy typically consists of a limited number of patients [16,17,18]. Previous research on advanced NSCLC with neoadjuvant treatment encompassed nearly all NSCLC subtypes, frequently in low-represented patient collectives, thus reducing the specificity of findings across diverse histologies [12,13,14,18,19,20,21,22,23,24,25,26,27]. Thus, this knowledge gap should be addressed in larger and specific patient collectives including lung adenocarcinoma (LUAD) and squamous cell carcinoma (SQCA) to deliver a more targeted and rigorous analysis.

By examining the perioperative parameters including laboratory tests, radiological and histological parameters, lung function, and clinical comorbidities, we aim to identify the predictors for the effectiveness of tumor regression in lung cancer patients undergoing neoadjuvant treatment and major lung cancer resections to improve the perioperative risk stratification and optimize patient management. The second goal of our study is to analyze the OS of the patients included in this study and find the relationship between survival and tumor regression.

## 2. Materials and Methods

### 2.1. Study Population

This single-institution retrospective cohort study was performed after approval by the Ethics Committee of the Ludwig Maximilian University of Munich (LMU), Germany, file number 24-0114, according to the Declaration of Helsinki and STROBE recommendations for clinical studies. Patients’ recruitment and treatment were performed at the Department of Thoracic Surgery of the Robert Bosch Hospital (Stuttgart, Germany) between 1 January 2016 and 31 December 2022. 

All patients with resectable malignant primary lung tumors undergoing neoadjuvant therapy and major surgical resections (lobectomy, bilobectomy, or pneumonectomy) with histologically reported lung adenocarcinoma (LUAD group) and squamous cell carcinoma (SQCA group) in the intraoperative histological specimens were included in the study. Patients undergoing similar treatment experiencing rare histological subtypes in the intraoperative specimens (e.g. adenosquamous carcinoma (*n* = 2), synovial sarcoma (*n =* 2), sarcomatoid lung carcinoma (*n =* 2), low-grade differentiated sarcoma of the lung (*n =* 1), neuroendocrine/small cell lung cancer (*n =* 2), and not otherwise specified tumors (NOS, *n* = 4)) were excluded (Figure 1). 

### 2.2. Data Assessments/Sources

Clinical data were collected from patients’ files stored in the hospital and the Onkostar database of the Baden-Wuerttemberg Cancer Registry. Primary lung tumors were categorized according to the 8th edition of the TNM staging system [28], and histopathological analysis was performed according to the World Health Organization Classification of lung tumors [29].

Clinical data included patients’ demographics (age, sex, BMI, smoking history, comorbidities), laboratory parameters collected 1-5 days before surgery (blood counts, C-reactive protein (CRP), creatinine, lactate dehydrogenase (LDH), and albumin), preoperative respiratory parameters (FVC: functional vital capacity; FEV1 forced expiratory volume in one second; DLCO: diffusing capacity of the lung for carbon monoxide), radiological (cTNM) parameters, and tumor and lymph node size (the largest distance in the transverse, coronal, and sagittal planes in CT imaging was used as the tumor and lymph node size). Histological parameters and pTNM were also collected. Data on patient treatment, including neoadjuvant and adjuvant regimens (chemotherapy, immunotherapy, radiation therapy), intraoperative approach (minimally invasive or open surgery), and postoperative tumor recurrence/metastasis and mortality, were collected.

### 2.3. Outcomes

The two outcomes of the study are the tumor regression and the survival of the lung cancer patients upon neoadjuvant treatment and major surgical resection. 

We therefore analyzed a broad spectrum of demographic data, as well as laboratory, lung functional, radiological, and histological parameters.

The correlation between these parameters and the intraoperative tumor regression score was analyzed concerning the histological subtype of the primary tumor (LUAD or SQCA). Tumor regression grade (TRG) was defined according to the initial study of Junker et al. (TRG_I: >95% vital tumor cells, TRG_IIa: >10% vital tumor cells, TRG_IIb: <10% vital tumor cells, TRG_III: absent vital tumor cells in the intraoperative histological specimens, respectively) [10,11]. TRG_III was also defined as complete pathologic response (CPR) according to Travis et al. [9].

### 2.4. Data Analysis

Continuous variables are presented as median and interquartile range. Comparisons between groups were performed using the Mann–Whitney U-test for continuous variables, and Pearson Chi-square test and Fisher’s exact test were used for binary variables. Tumor size reduction values after neoadjuvant treatment were given as absolute values (absolute delta value = tumor size before neoadjuvant treatment − tumor size after neoadjuvant treatment). To assess the percentual decrease of the tumor size, relative values were calculated as follows: (relative delta value = ((tumor size before neoadjuvant treatment − tumor size after neoadjuvant treatment)/tumor size before neoadjuvant treatment) × 100), values given in %. Multivariable analysis was performed by binary logistic regression analysis validated by three methods (Enter, Forward LR, and Backward LR). Odds ratios (OR) with 95% confidence intervals (CI) were assessed for independent predictors. To assess easily available parameters for clinical routine, optimal cut-off values derived by receiver-operator characteristics (ROC) and the Youden criterion were computed as binary variables and incorporated into the multivariable analysis. Survival data were generated by Kaplan Meier analysis (log-rank test) and the independent predictive value of the significant variables by Cox proportional hazard regression analysis. Overall survival is defined as the time interval between lung cancer surgery and event (death) or census (last recorded follow-up). Analyses were performed after excluding the missing values (under 10% for selected variables), by using SPSS (Version 26, IBM, Armonk, NY, USA). *p*-values < 0.05 were considered statistically significant.

## 3. Results

### 3.1. Study Population

Of 892 patients admitted for lung cancer surgery, 95 patients fulfilled the inclusion criteria. The analyzed cohort (33.7% female patients) included patients with a median age of 64.20 [57.90; 69.06] years and a median BMI of 24.54 [22.16; 27.04] kg/m^2^.

Based on the histology of the primary tumor, 58 patients (61.1%) and 37 patients (38.9%) were included in the LUAD and SQCA groups, respectively. 

The inclusion process and groups of patients categorized by the histology of the primary tumor are illustrated in Figure 1.

The LUAD group comprised significantly more female patients (27/58 (46.6%) vs. 5/37 (13.5%, *p =* 0.0009)), overweight patients (29/58 (50.0%) vs. 10/37 (27.0%), *p =* 0.0264)), as well as smokers reporting a lower nicotine consumption (30.0 [4.25; 45.0] vs. 40.0 [22.5; 52.5], *p =* 0.0288). 

No significant differences in the distribution of patients’ comorbidities were observed between groups.

Patients with SQCA were admitted with significantly lower lung function parameters in comparison to the LUAD patients (FEV1: 69.00 [55.50–85.00] vs 80.50 [67.00–90.25] % predicted, *p =* 0.0103; DLCO: 48.00 [42.00–62.00] vs 62.50 [50.50–70.50] % predicted, *p =* 0.0023 and Tiffeneau Index 88.00 [78.00–95.00] vs 95.00 [87.00–102.00] %, *p =* 0.0014, respectively). The patients’ demographics are summarized in Table 1.

Laboratory parameters showed a low hemoglobin level in all patients upon neoadjuvant therapy without significant differences between groups. Patients from the LUAD group were admitted with lower serum C-reactive protein levels (0.20 [0.10; 0.60] vs. 0.50 [0.10; 1.30] mg/dL, *p =* 0.0206) and higher serum albumin levels (4.27 [3.90; 4.46] vs. 4.00 [3.60; 4.35] g/dL, *p =* 0.0281). The laboratory parameters in both groups are presented in Appendix A.

No significant differences regarding tumor side, localization, and lymph node involvement were reported between groups. The LUAD group comprised significantly more patients with cT1 tumors (11/58 (19.0%) vs. 0/37 (0%), *p =* 0.0048) and fewer patients with cT4 tumors (23/58 (39.7%) vs. 24/37 (64.9%), *p =* 0.0166, Table 2).

The characteristics of the surgical procedures are presented in Table 3.

No significant differences on the tumor resection approach (*p =* 0.1939) were reported. Whereas the LUAD group included more patients undergoing lobectomies upon neoadjuvant treatment (50/58 (86.2%) vs. 25/37 (67.6%), *p =* 0.0298), the SQCA group comprised more patients undergoing pneumonectomies (8/37 (21.6%) vs. 4/58 (6.9%), *p=* 0.0351). 

Regarding the neoadjuvant regimen, 18 patients (18.9%) underwent chemotherapy alone, 7 patients (7.4%) chemoimmunotherapy, 1 patient (1.1%) radiation therapy, 64 patients (67.4%) chemoradiation, and 5 patients (5.3%) a combination of all three approaches (used drugs listed in Appendix A). Patients with combined neoadjuvant chemo- and radiation therapy were significantly more frequent in the SQCA group (30/37 (81.1%) vs. 34/58 (58.6%), *p = 0.0228*, Table 3).

The median in-hospital stay was significantly longer in patients with SQCA (13.0 [9.5; 24.5] vs. 11.0 [8.0; 14.0] days, *p =* 0.0458). Proportionally, the stay in the intensive care unit was longer in the SQCA group (3.0 [1.0; 5.0] vs. 1.0 [1.0; 2.0] days, *p =* 0.0039). 

The analysis of the intraoperative histological specimens revealed significantly more patients experiencing > 10% vital tumor cells (TRG_IIa) in the LUAD group (27/58 (46.6%) vs. 5/37 (13.5%), *p =* 0.0009) and more patients with no vital tumor cells (TRG_III) in the SQCA group (20/37 (54.1%) vs. 10/58 (17.2%), *p =* 0.0002). This was under the significantly higher incidence of yT0 tumors in the SQCA group (22/37 (59.5%) vs. 10/58 (17.2%), *p <* 0.0001) and lower incidence of yT4 tumors in the SQCA group (1/37 (2.7%) vs. 8/58 (13.8%), *p =* 0.0719, Table 4).

### 3.2. Logistic Regression Analysis of Risk Factors

The primary outcome of the study is the analysis of tumor regression of lung cancer patients undergoing neoadjuvant treatment and major surgical resection. To identify meaningful parameters that independently predict tumor regression, a binary regression analysis considering TRG I_II and TRG_III was performed on the histology of the primary tumor (LUAD, SQCA). Accordingly, the following parameters were confirmed as statistically significant: intraoperative histology (SQCA), lymph node size > 1.7 cm at the time of diagnosis, and absolute delta tumor size values upon neoadjuvant treatment > 2.6 cm. The logistic regression model is summarized in Table 5.

To assess easily available parameters for clinical routine, relative delta values indicating the percentage of tumor regression upon neoadjuvant therapy were calculated. Specifically, a relative delta > 30% was identified as an independent predictor of CPR. The binary logistic regression analysis was reproduced as sensitivity analysis by using relative delta values > 30% instead of absolute delta values (>2.6 cm), with qualitatively unchanged results (Appendix A).

### 3.3. Survival Analysis

The secondary outcome of the study is the analysis of overall survival in lung cancer patients undergoing neoadjuvant treatment and major surgical resection.

Median overall survival for the whole cohort was 71 [47.0–95.0] months. Forty-six death events were recorded in the whole cohort during 330 cumulative follow-up years (Figure 2A).

Patients aged 70 years or older experienced a reduced OS (*n* = 21, 5-year OS = 38.1%) in comparison to younger patients (*n* = 74, 5-year OS = 55.9%, *p =* 0.0325, Figure 2B).

No significant difference was noted in the OS of patients when considering BMI, smoking status, associated comorbidities, lung function, or laboratory parameters. Patients suffering from SQCA had slightly better OS (*n* = 37, 5-year OS = 53.5%) when compared to LUAD patients with no significant difference (*n* = 58, 5-year OS = 49.4%, *p =* 0.5041, Figure 2C). 

Tumor side, lobe localization, and surgical approach (open vs. minimally invasive) were not significantly associated with OS. An extended resection (>1 lobe, *n* = 20) was significantly associated with a worse OS (5-year OS = 30.0%) in comparison to patients undergoing standard lobectomies (*n* = 75, 5-year OS = 58.3%, *p =* 0.0070, Figure 2D).

A tumor stage > yT3 (*n* = 20) had a significantly decreased OS (5-year OS = 26.8%) in comparison to lower tumor stages (<yT3, *n* = 75, 5-year OS = 60.1%, *p =* 0.0251, Figure 2E). The lymph node involvement after neoadjuvant therapy was not significantly associated with OS. 

No significant differences in OS were reported when considering tumor regression grade (Figure 3A).

An absolute tumor size reduction (absolute delta) > 2.6 cm after neoadjuvant therapy was significantly associated with an improved OS (*n* = 24, 5-year OS = 81.8% vs. *n* = 71, 5-year OS = 43.1%, *p =* 0.0122, Figure 3B). Similarly, an improved OS was observed in the patient group experiencing a relative tumor size reduction (relative delta) > 30% (*n* = 52, 5-year OS = 65.1% vs. *n* = 43, 5-year OS = 38.8%, *p =* 0.0415).

Patients experiencing a tumor recurrence or metastasis during follow-up had a significantly worse OS in comparison to those patients without (*n* = 39, 5-year OS = 26.0% vs. *n* = 54, 5-year OS = 77.3%, *p =* 0.0017, Figure 3C).

To assess the independent predictive value of the abovementioned parameters on OS, a multivariate Cox regression analysis was performed. Accordingly, we identified age (>70 years), extended resections (>1 lobe), and tumor recurrence or metastasis during follow-up as independent negative predictors of long-term OS. These predictors increased the risk of death by 2.70-, 2.11-, and 2.41-fold, respectively. On the contrary, the tumor size reduction after neoadjuvant therapy (absolute delta > 2.6 cm) was an independent parameter of improved OS (Exp(B) 3.82 [1.33–10.92], *p =* 0.0126. Figure 3D, Table 6).

## 4. Discussion

The findings of our study shed light on the predictive value of histological and radiological parameters, including primary tumor histology, lymph node size, and tumor size upon neoadjuvant treatment initiation, in determining the effectiveness of tumor regression in patients undergoing major lung cancer resections. These insights have significant implications for treatment planning, prognostic assessment, and therapeutic decision making in the management of advanced lung cancer.

In our cohort, 30 patients (31.57%) experienced CPR upon neoadjuvant therapy. To date, the estimation of CPR in the preoperative setting is inexact. In the published data, the CPR incidence rate is variable, varying from 4% to 33% in various protocols [30,31,32].

Histological assessment of the primary tumor emerges as a critical determinant of treatment response in patients undergoing neoadjuvant therapy. Interestingly, CPR was significantly higher in the SQCA group. In contrast to our results, Zens et al. showed that the major pathological response after neoadjuvant treatment was more frequently observed in lung adenocarcinoma patients, where only 11% of SQCA had a complete pathological response [14]. In the retrospective analyses from Schreiner et al., CPR was also more observed in patients with adenocarcinoma [20].

Furthermore, various anatomical measurements of changes in tumor size and lymph node size on CT or PET/CT scans were analyzed to improve the preoperative CPR prediction [33]. A CT scan delivers anatomical information based on morphological tumor alterations. However, even after dramatic tumor downstaging, vital cells may still be present. Therefore, the morphological evaluation may be misleading in some patients. Despite this, we identified the rate of reduction in the primary tumor size on CT or PET/CT scans as an independent predictor of CPR. This result is in line with existing literature [34,35]. In contrast, no correlation between CT-based volume reduction and pathologic response was identified by Cerfolio et al. and Pöttgen et al. [36,37]. The inconsistency might have resulted from the small patient size of these studies. Moreover, we also identified lymph node size > 1.7 cm at the time of diagnosis as an independent predictor of CPR. Coroller et al. could also identify radiological features of lymph nodes as predictors of CPR after neoadjuvant treatment in NSCLC patients [38].

Previous studies assessing a similar treatment approach, involving neoadjuvant therapy followed by surgery, have reported a median OS ranging from 24 to 66 months [14,16,17,27,39]. In comparison, the median OS of 71 months observed in our series is encouraging. However, it should be emphasized that, in our cohort, patients receiving neoadjuvant therapy had five different neoadjuvant regimens. This is relevant because the abovementioned studies included only patients undergoing chemotherapy or chemoradiation therapy.

In our study, OS estimates were significantly reduced in patients aged over 70 years, in resections of more than one lobe group, and in patients experiencing recurrence or metastasis during the follow-up period. Pilotto et al. developed a scoring system for squamous cell carcinoma of the lung patients, where the age of patients was also one of the risk factors in predicting OS in adjuvant and neoadjuvant therapy groups [40]. On the other hand, Früh et al. showed a better OS in younger patients after neoadjuvant therapy and surgery [27]. The worse predictive value of the extensive lung resection and pneumonectomy in OS after neoadjuvant treatment was also confirmed in several studies [16,17,32,41]. The reduction in OS after confirmed metastasis or tumor progress is an already known phenomenon [27,42,43], which we also confirmed. The OS of patients experiencing > yT3 was also reduced; however, this finding was not significant in multivariable analysis. The same results were reported by Counago et al. [17]. 

Patients with a tumor size reduction of more than 2.6 cm after neoadjuvant therapy could show better OS in our cohort. In contrast to our findings, Tanahashi et al. could not find any survival benefits in patients with tumor size reduction after neoadjuvant therapy [34]. This could be explained by the difference in study population heterogeneity, cohort size, and chosen therapy regimens. In our cohort, patients suffering from squamous cell carcinoma had slightly better 5-year OS (53.5%) when compared to adenocarcinoma patients’ 5-year OS (49.4%). This result was not significant in multivariable analysis; however, it can still be explained by a high number of patients with CPR in the SQCA group. 

Surprisingly, in our cohort, we could not identify any significant benefit of OS in patients with pathological complete response in comparison to the CheckMate 816 and Keynote 671 studies [21,22]. It can also be explained by the heterogeneity of patient populations and treatment regimens. Another reason for this may be the shorter follow-up periods in the abovementioned studies. Moreover, we found no association between survival and lymph node involvement after neoadjuvant therapy and surgical resection, in contrast to Yang et al. and Counago et al.’s findings [17,41]. This could be attributed to the limited number of patients, the varying therapy regimens applied, and the differing follow-up durations across these studies.

While our study provides valuable insights into the predictive factors of tumor regression effectiveness in neoadjuvant-treated lung cancer patients, several limitations warrant consideration. The retrospective nature of our analysis may introduce inherent biases and confounding variables that could impact the interpretation of results. Additionally, the heterogeneity of patient populations and treatment regimens may limit the generalizability of our findings.

## 5. Conclusions

In summary, our study elucidates the predictive value of histological parameters, including primary tumor histology, lymph node involvement, and tumor size upon neoadjuvant treatment initiation, in determining the effectiveness of tumor regression in patients undergoing major lung cancer resections. These findings have significant implications for personalized treatment planning and prognostic assessment in the management of advanced lung cancer, ultimately guiding therapeutic decisions and improving patient outcomes.

Prospective studies incorporating larger patient cohorts and standardized treatment protocols are needed to validate the prognostic utility of these histological parameters and refine predictive models for treatment response in neoadjuvant-treated lung cancer patients.

## Figures and Tables

**Figure 1 cancers-16-02885-f001:**
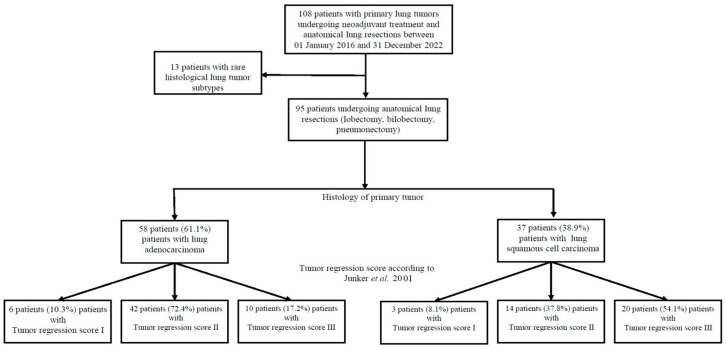
Study flow chart illustrating patient enrollment at study entry. Of 892 patients undergoing oncological thoracic surgery between 1 January 2016 and 31 December 2022, 108 (12.1%) patients with resectable primary lung tumors underwent neoadjuvant treatment and primary lung cancer resections. Thirteen (12.04%) patients experiencing rare histological tumor subtypes (e.g. adenosquamous carcinoma, synovial sarcoma, sarcomatoid carcinoma, low-grade differentiated sarcoma of the lung, neuroendocrine/small-cell lung cancer, and not otherwise specified tumors) were excluded from the study; thus, 95 of 108 patients (87.96%) with primary resectable lung tumors of the lung were included. Based on the histology of the primary tumor, patients were categorized into two groups: a lung adenocarcinoma group (LUAD, 58 patients, 61.1%), and a squamous cell carcinoma group (SQCA, 37 patients, 38.9%). Based on the tumor regression grade (TRG) of the intraoperative specimens, patients were further stratified in TRG_I (LUAD: 10.3%, SQCA: 8.1%), TRG_II (LUAD: 72.4%, SQCA: 37.8%), and TRG_III (LUAD: 17.2%, SQCA: 54.1%), respectively [11].

**Figure 2 cancers-16-02885-f002:**
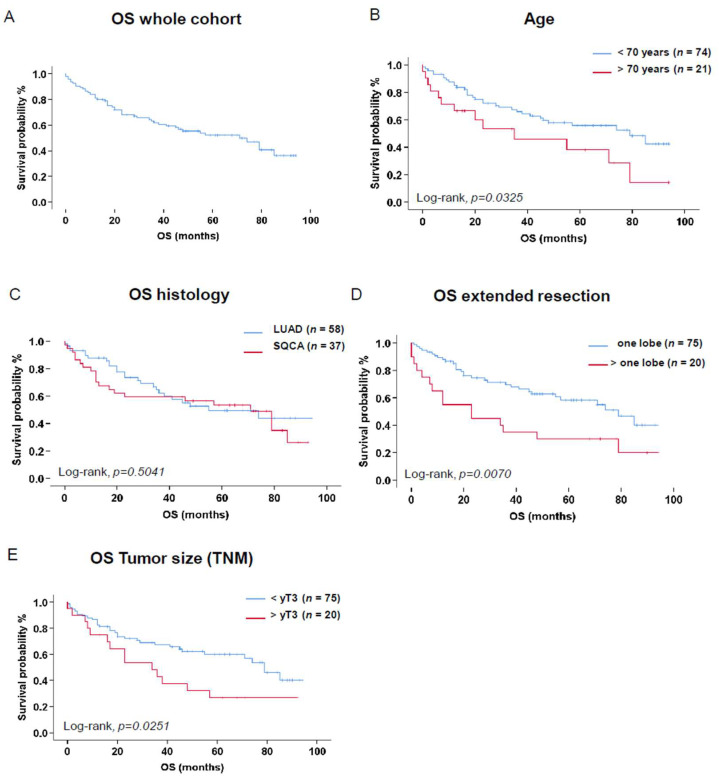
Kaplan–Meier survival analysis including patients at risk, reported events (death), and patients censored (25 January 2024) addressing the whole cohort (**A**), age > 70 years (**B**), histology of the primary tumor (LUAD vs. SQCA) (**C**), extended resections (>one lobe) (**D**), and tumor size upon neoadjuvant therapy (>yT3) (**E**). Comparison of the survival estimates was analyzed by log-rank test. *p*-values < 0.05 were considered significant. Abbreviations: OS: overall survival; LUAD: lung adenocarcinoma; SQCA: squamous cell lung cancer; TNM: tumor node metastasis staging system [28].

**Figure 3 cancers-16-02885-f003:**
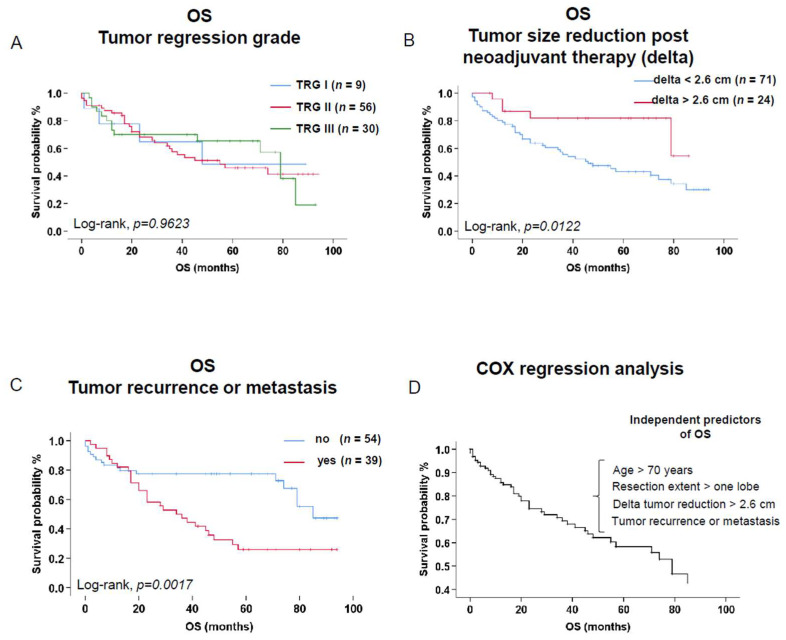
Kaplan–Meier survival analysis including patients at risk, reported events (death), and patients censored (25 January 2024) addressing the tumor regression grade (TRG I, II, and III) (**A**), tumor reduction size upon neoadjuvant therapy (absolute delta value > 2.6 cm) (**B**), and tumor recurrence or metastasis (**C**). Cox regression analysis including the independent predictors of OS (**D**). Comparison of the survival estimates was analyzed by log-rank test. *p*-values < 0.05 were considered significant. Abbreviations: OS: overall survival, TRG: tumor regression grade according to Junker et al. [11], absolute delta value = tumor size before neoadjuvant treatment − tumor size after neoadjuvant treatment.

**Table 1 cancers-16-02885-t001:** Demographics of patients undergoing neoadjuvant therapy and surgical resection classified by histology of primary tumor.

Patient Demographics at Study Entry	LUAD *n* = 58	SQCA *n* = 37	*p*-Value
**Age (median, quartiles [1st; 3rd])** years	60.73 [54.5; 69.0]	65.6 [60.5; 70.3]	0.0687
Age > 70 (*n*, %)	11/58 (19.0%)	10/37 (27.0%)	0.3558
**Sex (*n*, %)**			0.0009
Female	27/58 (46.6%)	5/37 (13.5%)	
Male	31/58 (53.4%)	32/37 (86.5%)	
**BMI (median, quartiles [1st; 3rd])**	25.3 [23.0; 27.2]	23.8 [21.8; 26.6]	0.1842
BMI > 18.5, <25 kg/m^2^ (*n*, %)	25/58 (43.1%)	24/37 (64.9%)	0.0385
BMI > 25, <30 kg/m^2^ (*n*, %)	29/58 (50.0%)	10/37 (27.0%)	0.0264
**Pack years (median, quartiles [1st; 3rd])**	30.0 [4.25; 45.0] PY	40.0 [22.5; 52.5] PY	0.0288
Never smokers (*n*, %)	10/58 (17.2%)	3/37 (8.1%)	0.2066
Current smokers (*n*, %)	32/58 (55.2%)	18/37 (48.6%)	0.5346
Ex-smokers (*n*, %)	16/58 (27.6%)	16/37 (43.2%)	0.1143
**Comorbidities (*n*, %)**			
Respiratory	21/58 (36.2%)	18/37 (48.6%)	0.4546
Cardiovascular	9/58 (15.5%)	2/37 (5.4%)	0.1331
Renal	2/58 (3.4%)	3/37 (8.1%)	0.3744
Liver	3/57 (5.3%)	3/37 (8.1%)	0.6769
Neurological/psychiatric	8/58 (13.8%)	3/37 (8.1%)	0.3984
Diabetes mellitus	5/58 (8.6%)	7/37 (18.9%)	0.1407
Non-pulmonary malignancies	12/58 (20.7%)	4/37 (10.8%)	0.2096
**Lung function parameters (median, quartiles [1st; 3rd])**			
FVC (predicted, %)	82.50 [72.75–98.00]	79.00 [71.50–91.00]	0.2903
FEV1 (predicted, %)	80.50 [67.00–90.25]	69.00 [55.50–85.00]	0.0103
DLCO (predicted, %)	62.50 [50.50–70.50]	48.00 [42.00–62.00]	0.0023
FEV1/FVC (%)	95.00 [87.00–102.00]	88.00 [78.00–95.00]	0.0014

For continuous variables, a non-parametric Mann-Whitney U test was performed. For binary variables, the Pearson Chi-square test or Fisher’s exact test was performed. *p*-values < 0.05 are statistically significant. Abbreviations: LUAD: lung adenocarcinoma; SQCA: squamous cell lung cancer; BMI: body mass index; PY: pack years; FVC: functional vital capacity; FEV1: forced expiratory volume in one second; DLCO: diffusing capacity of the lung for carbon monoxide.

**Table 2 cancers-16-02885-t002:** Tumor characteristics in patients undergoing neoadjuvant therapy and surgical resection classified by histology of primary tumor.

Tumor Characteristics	LUAD *n* = 58	SQCA *n* = 37	*p*-Value
**Tumor side (*n*, %)**			
Left	27/58 (46.6%)	16/37 (43.2%)	0.7521
Right	31/58 (53.4%)	21/37 (56.8%)	
**TNM8 classification (*n*, %)**			
cT1	11/58 (19.0%)	0/37 (0%)	0.0048
cT2	10/58 (17.2%)	4/37 (10.8%)	0.3886
cT3	14/58 (24.1%)	9/37 (24.3%)	0.9830
cT4	23/58 (39.7%)	24/37 (64.9%)	0.0166
**Lymph node involvement (*n*, %)**			
cN0	9/58 (15.5%)	4/37 (10.8%)	0.5151
cN1	11/58 (19.0%)	7/37 (18.9%)	0.9955
cN2	35/58 (60.3%)	22/37 (59.5%)	0.9315
cN3	3/58 (5.2%)	4/37 (10.8%)	0.3050

For binary variables, the Pearson Chi-square test or Fisher’s exact test was performed. *p*-values < 0.05 are statistically significant. Abbreviations: LUAD: lung adenocarcinoma; SQCA: squamous cell lung cancer; cT1–4: tumor stage 1–4 according to the TNM8 (tumor node metastasis staging system 8 [28]) assessed on computed tomography; cN0-3: lymph node involvement according to the TNM8 on computed tomography.

**Table 3 cancers-16-02885-t003:** Technical aspects of the tumor resection in patients undergoing neoadjuvant therapy and surgical resection classified by histology of the primary tumor.

Features of the Surgical Procedures	LUAD *n* = 58	SQCA *n* = 37	*p*-Value
**Surgical approach (*n*, %)**			
Open (thoracotomy)	50/58 (86.2%)	35/37 (94.6%)	0.1939
Minimally invasive (VATS)	5/58 (8.6%)	2/37 (5.4%)	0.7017
Conversion to open	3/58 (5.2%)	0/37 (0.0%)	0.2791
**Resection extent (*n*, %)**			
Lobectomy	50/58 (86.2%)	25/37 (67.6%)	0.0298
Bilobectomy	4/58 (6.9%)	4/37 (10.8%)	0.7071
Pneumonectomy	4/58 (6.9%)	8/37 (21.6%)	0.0351
**Neoadjuvant therapy**			
Chemotherapy	14/58 (24.1%)	4/37 (10.8%)	0.1060
Chemo- and immunotherapy	5/58 (8.6%)	2/37 (5.4%)	0.7017
Radiation therapy	0/58 (0.0%)	1/37 (2.7%)	0.3895
Chemo- and radiation therapy	34/58 (58.6%)	30/37 (81.1%)	0.0228
Chemo-, immuno-, and radiation therapy	5/58 (8.6%)	0/37 (0.0%)	0.1527
**Adjuvant therapy**			
Chemotherapy	1/57 (1.8%)	0/36 (0.0%)	1.0000
Immunotherapy	9/57 (15.8%)	0/36 (0.0%)	0.0121
Chemo- and immunotherapy	0/57 (0.0%)	1/36 (2.8%)	0.3871
Radiation therapy	8/57 (14.0%)	1/36 (2.8%)	0.0737
Chemo-, immune-, and radiation therapy	1/57 (1.8%)	1/36 (2.8%)	1.0000
**Primary tumor relapse or metastasis (*n*, %)**	31/57 (54.4%)	8/36 (22.2%)	0.0022
**Mortality (*n*, %)**			
During maximal follow-up	25/58 (43.1%)	21/37 (56.8%)	0.5041
**Overall survival (estimate [lower bound** **; upper bound], months)**	55.00 [17.06; 92.94]	71.00 [44.89; 97.11]	0.5041

For continuous variables, a non-parametric Mann–Whitney U test was performed. For binary variables, the Pearson Chi-square test or Fisher’s exact test was performed. *p*-values < 0.05 are statistically significant. Abbreviations: LUAD: lung adenocarcinoma; SQCA: squamous cell lung cancer; VATS: video-assisted thoracoscopic surgery.

**Table 4 cancers-16-02885-t004:** Characterization of tumor regression proportion score according to Junker et al. [11] in patients undergoing neoadjuvant therapy and major surgical resections classified by histology of the primary tumor.

Characterization of Tumor Regression Score	LUAD *n* = 58	SQCA *n* = 37	*p*-Value
**TNM8 classification (*n*, %)**			
yT0	10/58 (17.2%)	22/37 (59.5%)	<0.0001
yT1	20/58 (34.5%)	10/37 (27.0%)	0.4459
yT2	11/58 (19.0%)	2/37 (5.4%)	0.0608
yT3	9/58 (15.5%)	2/37 (5.4%)	0.1331
yT4	8/58 (13.8%)	1/37 (2.7%)	0.0719
**Lymph node involvement (*n*, %)**			
yN0	34/58 (58.6%)	33/37 (89.2%)	0.0014
yN1	6/58 (10.3%)	4/37 (10.8%)	0.9425
yN2	17/58 (29.3%)	0/37 (0.0%)	0.0003
yN3	1/58 (1.7%)	0/37 (0.0%)	1.0000
**Tumor regression grade (*n*, %)**			
TRG_I (>95% vital tumor cells)	6/58 (10.3%)	3/37 (8.1%)	0.7166
TRG_IIa (>10% vital tumor cells)	27/58 (46.6%)	5/37 (13.5%)	0.0009
TRG_IIb (<10% vital tumor cells)	15/58 (25.9%)	9/37 (24.3%)	0.8664
TRG_III (CPR, no vital tumor cells)	10/58 (17.2%)	20/37 (54.1%)	0.0002
**Tumor size in CT (median, quartiles [1st; 3rd]) (cm)**			
Before neoadjuvant treatment	3.95 [2.40; 6.68]	5.50 [3.60; 6.35]	0.2105
After neoadjuvant treatment	2.55 [1.50; 4.60]	3.30 [1.85; 4.35]	0.3758
Absolute delta value	1.15 [0.55; 2.50]	1.70 [0.60; 3.10]	0.4248
Relative delta value	30.73 [14.95; 53.55]	32.14 [14.17; 51.05]	0.8397
**Lymph node size in CT (median, quartiles [1st; 3rd]) (cm)**			
Before neoadjuvant treatment	1.60 [1.20; 2.03]	1.60 [1.00; 2.20]	0.9299

For continuous variables, a non-parametric Mann–Whitney U test was performed. For binary variables, the Pearson Chi-square test or Fisher’s exact test was performed. *p*-values < 0.05 are statistically significant. Abbreviations: LUAD: lung adenocarcinoma; SQCA: squamous cell lung cancer; yT0–4: tumor stage 0–4 according to the TNM8 [28] staging system upon neoadjuvant therapy assessed from the intraoperative histology; yN0–3: lymph node involvement according to the TNM8 upon neoadjuvant therapy assessed from the intraoperative histology; TRG: tumor regression grade according to the classification of Junker et al. [11]; CPR: complete pathological response; absolute delta value = tumor size before neoadjuvant treatment − tumor size after neoadjuvant treatment; relative delta value = ((tumor size before neoadjuvant treatment − tumor size after neoadjuvant treatment)/tumor size before neoadjuvant treatment) × 100.

**Table 5 cancers-16-02885-t005:** Binary logistic regression model predicting complete pathologic response in primary lung cancer patients undergoing neoadjuvant therapy and major surgical resections.

Covariates for Tumor Regression	Exp(B) [95% CI]	*p*-Value
Intraoperative histology (LUAD vs SQCA)	6.88 [2.40–19.77]	0.0003
Lymph node size in pre-neoadjuvant PET > 1.7 cm	3.13 [1.11–8.83]	0.0310
Absolute delta value > 2.6 cm	3.76 [1.20–11.81]	0.0233

Abbreviations: LUAD: lung adenocarcinoma; SQCA: squamous cell lung cancer; PET: positron emission computed tomography; Exp(B) = odds ratio, 95% confidence interval [lower bound–upper bound]. *p*-values < 0.05 are statistically significant. Absolute delta value = tumor size before neoadjuvant treatment − tumor size after neoadjuvant treatment.

**Table 6 cancers-16-02885-t006:** Cox proportional hazard regression analysis predicting overall survival in primary lung cancer patients undergoing neoadjuvant therapy and major surgical resections.

Independent Predictors of Overall Survival	Exp(B) [95% CI]	*p*-Value
Age (>70 years)	2.70 [1.37–5.36]	0.0043
Extended resections (>one lobe)	2.11 [1.10–4.08]	0.0257
Absolute delta value > 2.6 cm	3.82 [1.33–10.92]	0.0126
Tumor recurrence or metastasis	2.41 [1.27–4.54]	0.0068

Abbreviations: Exp(B) = hazard ratio, 95% confidence interval [lower bound–upper bound]. *p*-values < 0.05 are statistically significant. Absolute delta value = tumor size before neoadjuvant treatment − tumor size after neoadjuvant treatment.

## Data Availability

The datasets of the current study are available from the corresponding author upon reasonable request.

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
