# Peer review of "Perioperative Predictive Factors for Tumor Regression and Survival in Non-Small Cell Lung Cancer Patients Undergoing Neoadjuvant Treatment and Lung Resection"

_cancers, 2024, doi:10.3390/cancers16162885_

Round 1

Reviewer 1 Report

Comments and Suggestions for Authors

I would like to congratulate the author of the interesting article titled “Histology of the primary tumor, lymph node, and tumor size upon neoadjuvant treatment predict effectiveness of complete pathologic response in patients undergoing primary lung cancer resections.”

The methodology used by the authors is appropriate, and the work is written in high-quality English. The sample size is sufficient, the study provides interesting results.

1. The current title is quite long and complicated. I suggest rephrasing it for clarity and conciseness.

2. The introduction should be slightly more extensive. It should analyze the current state of knowledge in more detail, identify knowledge gaps, and define the objectives of the study.

3. Why were lung tumors categorized according to the 7th edition instead of the 8th edition of the TNM staging system?

4. The endpoints (outcomes) of the study should be clearly and unambiguously specified.

5. The results section is too extensive, making it difficult for first-time readers to comprehend and analyze. Most of the results do not correspond to the analysis of the adopted outcomes. I propose significantly shortening the results section, placing less important data in tables or supplementary materials, and emphasizing the results that are crucial to the study.

6. In the discussion, the authors should conduct a more thorough review of the literature and relate the results obtained to this literature.

In summary, while the study is well-conducted and provides some valuable insights, it requires significant revisions to improve clarity and relevance.

Comments on the Quality of English Language

No comments

Author Response

Reviewer 1

I would like to congratulate the author of the interesting article titled “Histology of the primary tumor, lymph node, and tumor size upon neoadjuvant treatment predict the effectiveness of complete pathologic response in patients undergoing primary lung cancer resections.”The methodology used by the authors is appropriate, and the work is written in high-quality English. The sample size is sufficient, the study provides interesting results.

Comment 1:  The current title is quite long and complicated. I suggest rephrasing it for clarity and conciseness.
Answer 1:  Thank you for pointing this out.  We agree with the reviewer that the title is long and complicated.

Change 1: In the new version of the manuscript, we provided a simplified title as follows:

“Perioperative predictive factors for tumor regression and survival in Non-small cell lung cancer patients undergoing neoadjuvant treatment and lung resection.”

Comment 2: The introduction should be slightly more extensive. It should analyze the current state of knowledge in more detail, identify knowledge gaps, and define the objectives of the study.

Answer 2: Thank you for highlighting this aspect. We agree with the reviewer and accordingly made changes in the Introduction section.

Changes 2: We added and addressed the current state of knowledge and also updated the reference list (lines 63-71, 75-77, 80-86).

Comment 3: Why were lung tumors categorized according to the 7th edition instead of the 8th edition of the TNM staging system?

Answer 3: Thank you for pointing this out. We apologize for this mistake. All patients included in the study were categorized by the 8th edition of the TNM system.

Changes 3: In the new version of the manuscript we revised this mistake and provided accordingly appropriate references.

Comment 4: The endpoints (outcomes) of the study should be clearly and unambiguously specified.

Answer 4: Thank you for pointing out this important aspect. We agree with the reviewer that the outcomes of the study were not clearly presented in the first version of the manuscript. The primary outcomes of the study were: tumor regression and the survival of the lung cancer patients upon neoadjuvant treatment. We accordingly aimed to analyze the perioperative predictors that might be associated with tumor regression and survival. For this reason, we analyzed a broad spectrum of demographic data, as well as laboratory, lung functional, radiological, and histological parameters that are routinely performed before surgery.

Changes 4: Following the reviewer’s concern, we rephrased the Materials and Methods section Data assessments and Outcomes, by clearly highlighting the endpoints of the study. In addition, we specified the outcomes of the study in the Result section.

Comment 5: The results section is too extensive, making it difficult for first-time readers to comprehend and analyze. Most of the results do not correspond to the analysis of the adopted outcomes. I propose significantly shortening the results section, placing less important data in tables or supplementary materials, and emphasizing the results that are crucial to the study.

Answer 5: Thank you for this comment. As correctly observed by the reviewer, a comprehensive assessment of preoperative risk factors and demographics was presented in Tables 1, 2, and 3. We agree with the reviewer that a more concise presentation of relevant results must be performed to improve the readability of the manuscript.

For this reason, we consistently reduced the content of the tables, addressing relevant clinical, laboratory, lung functional, radiological, and histological data. Similarly, we consistently reduced the content in the Result section.

Changes 5: In Table 1 data on age, BMI, alcohol consumption, and redundant lung function given in liters were removed. Tables 2 and 7 were moved to the supplementary material ( now Supplementary Table 1 and Supplementary Table 2). From the previous Table 3 (now Table 2) we removed the data regarding the lobar distribution of the tumor, data on the involvement of the pleura visceralis as well as postoperative histological TNM classification. From the previous Table 4 (now Table 3) we removed the side of the tumor, data on topographical resection, data on surgery time, as well as data on in hospital stay and 30-day mortality information. The Result section was substantially reduced and accordingly rephrased.    

Comment 6: In the discussion, the authors should conduct a more thorough review of the literature and relate the results obtained to this literature. 

In summary, while the study is well-conducted and provides some valuable insights, it requires significant revisions to improve clarity and relevance.

Answer 6:  Thank you for highlighting this aspect. We agree with the reviewer and accordingly made changes in the Discussion section.

Changes 6: We conducted more reviews of the literature and related the results to the literature, and also updated the reference list (lines 371-377, 382-389, 394-397, and 402-406).

Reviewer 2 Report

Comments and Suggestions for Authors Histology of the primary tumor, lymph node, and tumor size upon
neoadjuvant treatment predict effectiveness of complete pathologic response in patients undergoing primary lung cancer resections, looks primarily at a tumor
regression metrics during neoadjuvant therapy followed by surgery as a prognostic
index. While some of the analysis seems of interest, I am a little confused over
presentation of some factors I would take for granted, like tumor recurrence and
metastasis, possibly even age. Are these shown to add some support for the rest
of the analysis? Minor point is the formatting of the Table columns that are out
of alignment.

Comments on the Quality of English Language

OK

Author Response

Reviewer 2

Histology of the primary tumor, lymph node, and tumor size upon neoadjuvant treatment predict effectiveness of complete pathologic response in patients undergoing primary lung cancer resections, looks primarily at a tumor regression metrics during neoadjuvant therapy followed by surgery as a prognostic index.

Comment 1: While some of the analysis seems of interest, I am a little confused over presentation of some factors I would take for granted, like tumor recurrence and metastasis, possibly even age. Are these shown to add some support for the rest of the analysis?

Answer 1: Thank you for this comment. We agree with the reviewer that the presentation of the results is very detailed, making the selection of the clinically relevant parameters difficult. In the first version of the manuscript, we included all clinical data, laboratory, lung functional, radiological, and histological parameters, to comprehensively characterize the clinical condition of the patients. Even though it is known that age> 70y, tumor recurrence, and metastasis are predictors for negative outcomes in lung cancer patients, we kept these clinically relevant parameters in the manuscript to validate and discuss our results with previously published data.

The abovementioned parameters were found by Kaplan-Meier analysis as predictors of negative survival, whereas their independent predictive values were further confirmed by multivariable Cox regression analysis. 

In addition, we consistently reduced the content of the tables, addressing relevant clinical, laboratory, lung functional, radiological, and histological data and, the content in the Result section.

Change 1: According to the reviewer’s suggestions, we substantially simplified the results, removing redundant or less meaningful parameters, to improve the clarity of the manuscript.

Comment 2: Minor point is the formatting of the Table columns that are out of alignment.

Answer and Change 2: Thank you very much. The table template is automatically generated by the submission system of the journal. We will contact the Editorial office of the Journal to address this issue.

Reviewer 3 Report

Comments and Suggestions for Authors

The present work is a retrospective study on patients who received neoadjuvant treatments for lung cancer; its aim was to identify factors predictive of pathological response. 

The work is quite confusing.

Too much tables are presented: 6, 7 and 8 might be unified while table 2 and 3 might be avoided.

Important data are lacking, as the chemo-regimens used in neoadjuvant setting; moreover, 1 patient received radiation therapy alone?

In general, the work lacks novelty. 

Comments on the Quality of English Language

Acceptable

Author Response

Reviewer 3

The present work is a retrospective study on patients who received neoadjuvant treatments for lung cancer; its aim was to identify factors predictive of pathological response.

Comment 1: The work is quite confusing. Too much tables are presented: 6, 7 and 8 might be unified while table 2 and 3 might be avoided.

Answer 1: Thank you for this comment. We agree with the reviewer that the presentation of the results is very detailed, making the selection of the clinically relevant parameters difficult. In the first version of the manuscript, we included all clinical data, laboratory, lung functional, radiological, and histological parameters, to comprehensively characterize the clinical condition of the patients.

Change 1: Following the reviewer’s concern we substantially reduced the content of the Result section to improve the clarity of the manuscript. In the updated version of the manuscript we:

  1. removed redundant sentences in the results section by keeping relevant data in the presented tables
  2. removed redundant or less meaningful parameters from Table 1 (age, BMI, alcohol consumption, and redundant lung function given in liters), Table 3 (lobar distribution of the tumor, data on the involvement of the visceral pleura as well as postoperative histological TNM classification) and Table 4 (the side of the tumor, data on topographical resection, data on surgery time, as well as data on in-hospital stay and 30-day mortality information)
  • moved the Tables 2 and 7 to the supplementary material as Supplementary Table 1 and Supplementary Table 3.

Since Table 6 (now Table 5) and 8 (now Table 6) present different outcomes and analyses (Table 6 - logistic regression model predicting complete pathologic response, and Table 8 - logistic regression model predicting overall survival), we deliberately kept both analyses in separate tables.  Table 7 was moved to the supplementary material, as suggested by the reviewer.

Comment 2: Important data are lacking, as the chemo-regimens used in neoadjuvant setting

Answer 2: Thank you very much for pointing this important aspect out. Indeed, the initial version of the manuscript did not include data on chemotherapy substances and combined regimens. Since our study covers a period of 7 years, 10 different neoadjuvant regimens were administered according to the recommendations of our interdisciplinary tumor board.

Change 2: Following the reviewer’s recommendation, we included in the revised version of the manuscript an additional table with chemotherapy substances and combined regimens as supplementary material (Supplementary Table 2).

Comment 2: 1 patient received radiation therapy alone?

Answer 2:  The patient undergoing radiation therapy alone was a 78 y.o. male patient, with squamous cell carcinoma (cT4cN1cM0), in poor clinical condition, with cardiac and renal illness. Therefore, our interdisciplinary tumor board recommended radiation therapy alone (45Gy) before surgery. Accordingly, the histological report showed a major pathological response (<10% vital tumor cells) of the tumor after radiation therapy.

Comment 3: In general, the work lacks novelty.

Answer 3: Thank you for your constructive comment. The cohort undergoing neoadjuvant therapy is generally based on a limited number of patients. In our high-volume thoracic surgery department, around 10% of the surgically treated lung cancer patients underwent neoadjuvant therapy, in accordance with previously published studies [1, 2].

To comprehensively characterize this small-numbered patient collective, we aimed to provide a detailed analysis of all clinical data available for these patients. These parameters included laboratory, lung functional, radiological, and histological parameters. Even though, some low-sampled studies already addressed this topic [1, 3], some aspects regarding the histology of the primary tumor (lung adenocarcinoma vs. squamous cell carcinoma), as well as the tumor size reduction upon neoadjuvant treatment (absolute delta >2.6 cm, relative delta >30%) left insufficiently characterized.

Taking these considerations into account, the novelty of our study consists of the analysis of the histological subtypes of the primary tumor (lung adenocarcinoma vs. squamous cell carcinoma) in relation to the tumor regression grade after neoadjuvant therapy. In addition, we analyzed the independent predictive value of these parameters on survival.

References:

  1. Isobe, K., Y. Hata, S. Sakaguchi, F. Sato, S. Takahashi, K. Sato, G. Sano, K. Sugino, S. Sakamoto, Y. Takai, et al., Pathological response and prognosis of stage III non-small cell lung cancer patients treated with induction chemoradiation. Asia Pac J Clin Oncol, 2012. 8(3): p. 260-6.
  2. Schreiner, W., S. Gavrychenkova, W. Dudek, R.J. Rieker, S. Lettmaier, R. Fietkau, and H. Sirbu, Pathologic complete response after induction therapy-the role of surgery in stage IIIA/B locally advanced non-small cell lung cancer. J Thorac Dis, 2018. 10(5): p. 2795-2803.
  3. Counago, F., S. Montemuino, M. Martin, B. Taboada, P. Calvo-Crespo, M.P. Samper-Ots, P. Alcantara, J. Corona, J.L. Lopez-Guerra, M. Murcia-Mejia, et al., Prognostic factors in neoadjuvant treatment followed by surgery in stage IIIA-N2 non-small cell lung cancer: a multi-institutional study by the Oncologic Group for the Study of Lung Cancer (Spanish Radiation Oncology Society). Clin Transl Oncol, 2019. 21(6): p. 735-744.

Round 2

Reviewer 1 Report

Comments and Suggestions for Authors

Thank you for taking my comments on the article into account. I have no further comments.

Reviewer 2 Report

Comments and Suggestions for Authors

The authors responded to by concerns adequately.

Reviewer 3 Report

Comments and Suggestions for Authors

The authors sufficiently responded the questions. Originality and scientific soundness remain weak, however the article may be acceptable in the present form.